# A cost/benefit analysis of clinical trial designs for COVID-19 vaccine candidates

Donald A. Berry[1,2], Scott Berry[2], Peter Hale[3], Leah Isakov[4], Andrew W. Lo[5,6]*, Kien Wei Siah [6], Chi Heem Wong[6]

**1** Department of Biostatistics, University of Texas, Houston, Texas, United States of America, **2** Berry Consultants, Austin, Texas, United States of America, **3** The Foundation for Vaccine Research, Washington, D.C., United States of America, **4** Seqirus, Cambridge, Massachusetts, United States of America, **5** MIT Sloan School of Management, Cambridge, Massachusetts, United States of America, **6** Department of Electrical Engineering and Computer Science, Massachusetts Institute of Technology, Cambridge, Massachusetts, United States of America

* alo-admin@mit.edu

**Data Availability Statement:** The epidemiology data can be found at: https://github.com/CSSEGISandData/COVID-19.

## Abstract

We compare and contrast the expected duration and number of infections and deaths averted among several designs for clinical trials of COVID-19 vaccine candidates, including traditional and adaptive randomized clinical trials and human challenge trials. Using epidemiological models calibrated to the current pandemic, we simulate the time course of each clinical trial design for 756 unique combinations of parameters, allowing us to determine which trial design is most effective for a given scenario. A human challenge trial provides maximal net benefits—averting an additional 1.1M infections and 8,000 deaths in the U.S. compared to the next best clinical trial design—if its set-up time is short or the pandemic spreads slowly. In most of the other cases, an adaptive trial provides greater net benefits.

## 1 Introduction

The COVID-19 pandemic has caused the deaths of hundreds of thousands. Its economic fallout has upended the lives of billions, and caused trillions of dollars in financial losses. Life may not return to normal until a vaccine is found [1]. Despite the many candidates undergoing testing, an approved vaccine is not expected until 2021, even with substantially compressed development timelines [2], smooth proceeding of clinical trials, and not accounting for possible failures [3]. It is possible—though considered highly unlikely at the present time—that, like many non-influenza epidemics, the crisis may be over before a successful vaccine is developed [4].

Unlike typical therapeutics that are administered to sick patients, vaccines are intended for the healthy. Therefore, confirming the safety and effectiveness of a vaccine is of critical importance [5]. The two primary methods for demonstrating vaccine safety and efficacy are through either a vaccine efficacy randomized clinical trial (RCT) or a vaccine immunogenicity RCT. In the former, large numbers of recruited healthy volunteers are randomly selected to receive either the vaccine or a placebo/active control and then monitored for a period of time. At the end of the surveillance period, the difference in the proportion of infections between the

**Funding:** L.I. is an employee of the biotech company Seqirus, and D.A.B. and S.B are employees of Berry Consultants. These companies did not have any role in study design, data collection and analysis, decision to publish, or preparation of this manuscript. The specific roles of these authors are specified in the 'author contributions' section. Funding support from the MIT Laboratory for Financial Engineering is gratefully acknowledged, but no direct funding was received for this study.

**Competing interests:** P.H., K.S., and C.W. report no conflicts. L.I. is an employee of the biotech company Seqirus and receives salary and company stock as part of compensation. A.L. reports personal investments in private biotech companies, biotech venture capital funds, and mutual funds. A. L. is a co-founder and partner of QLS Advisors, a healthcare analytics and consulting company; an advisor to BrightEdge Ventures; a director of BridgeBio Pharma, Roivant Sciences, and Annual Reviews; chairman emeritus and senior advisor to AlphaSimplex Group; and a member of the Board of Overseers at Beth Israel Deaconess Medical Center and the NIH's National Center for Advancing Translational Sciences Advisory Council and Cures Acceleration Network Review Board. During the most recent six-year period, A.L. has received speaking/consulting fees, honoraria, or other forms of compensation from: AIG, AlphaSimplex Group, BIS, BridgeBio Pharma, Citigroup, Chicago Mercantile Exchange, Financial Times, FONDS Professionell, Harvard University, IMF, National Bank of Belgium, Q Group, Roivant Sciences, Scotia Bank, State Street Bank, University of Chicago, and Yale University. Funding support from the MIT Laboratory for Financial Engineering is gratefully acknowledged, but no direct funding was received for this study, no commercial funding was provided or solicited for this study, and no funding bodies had any role in study design, data collection and analysis, decision to publish, or preparation of this manuscript. The authors were personally salaried by their institutions during the period of writing (though no specific salary was set aside or given for the writing of this manuscript). This does not alter our adherence to PLOS ONE policies on sharing data and materials.

treatment and control arms is computed to demonstrate the ability of the vaccine to prevent infection or disease. A phase 3 vaccine efficacy RCT typically takes five to ten years to complete [6].

In a vaccine immunogenicity RCT, the primary endpoint is an immunity measurement or surrogate marker which is known to correlate with protection against infection or a disease. This type of trial involves a smaller number of volunteers and requires a shorter follow-up period, and as a result, is quicker and less costly [7]. Given that SARS-CoV-2 is a novel pathogen for which we do not yet know how to determine whether a subject is protected, vaccine efficacy must be confirmed using the longer and more costly vaccine efficacy RCT. While there exists the possibility of an expedited (conditional) licensure based on immunogenicity results with post-approval commitments, we find it unlikely to occur given the latest information. The U.S. Food and Drug Administration (FDA) has also issued a guidance stating that "the goal of development programs should be to pursue traditional approval via direct evidence of vaccine efficacy" [8].

A human challenge trial (HCT), in which volunteers are randomized into either the vaccine or placebo arm and then infected deliberately with live virus in a controlled setting, has been proposed as an alternative to accelerate the vaccine development process. Upon challenge, HCTs can quickly demonstrate safety and efficacy of candidate vaccines in preventing infection or disease. Depending on sample size, HCTs could also help to establish functional immune correlates of protection to inform the design of future vaccines. Since an HCT allows comparison of immune responses in vaccinated and unvaccinated individuals, precise measurements of post-vaccination viral loads, characterization of immune responses (innate, adaptive, cell-mediated) and antibody titers, and close monitoring and care of patients, it can help establish the correlates of protection and prove vaccine efficacy concurrently. Moreover, a properly designed HCT can determine transmission risk of the infected in a controlled setting with minimal exposure to investigators and the public. While concerns have been raised regarding the ethics and morality of HCTs, it is generally accepted that HCTs are ethically permissible when the benefits to society outweigh acknowledged risks [9], and they have been deemed acceptable for developing vaccines for multiple infectious diseases such as influenza [10], malaria [11], typhoid [12], cholera [13], and dengue fever [14]. To the best of our knowledge, there have been no published studies of a quantitative analysis of the potential societal value of a COVID-19 HCT.

In this paper, we compare the costs and benefits—as measured by the number of deaths and infections avoided—of confirming the safety and efficacy of a COVID-19 vaccine using four clinical trial designs: a traditional vaccine efficacy RCT, a vaccine efficacy RCT with an optimized surveillance period that maximizes the benefits of the trial (ORCT), an adaptive vaccine efficacy RCT (ARCT), and an HCT. Although our framework applies more broadly to any vaccine candidate for any infectious disease, we calibrate our simulations using a set of estimated epidemiological models for the SARS-CoV-2 virus (one for each of the 50 states and Washington, D.C.) to determine attack rates and cumulative numbers of infections and deaths in the U.S under various scenarios.

A summary of our simulation framework is displayed in Fig 1. We first estimate baseline models from data and make assumptions for the evolution of the epidemic in order to predict the attack rates over the course of the clinical trials. We then combine the attack rates with the assumptions for the vaccine trial design to simulate the outcomes for the clinical trials. Conditioned on the vaccine being approved, we make assumptions on the vaccination schedule and simulate the path of the epidemic in order to compute the net infections and deaths prevented.

Assuming that a clinical trial testing a vaccine with a true efficacy of 50% and using superiority tests starts on August 1, 2020, we estimate the date of licensure of the hypothetical

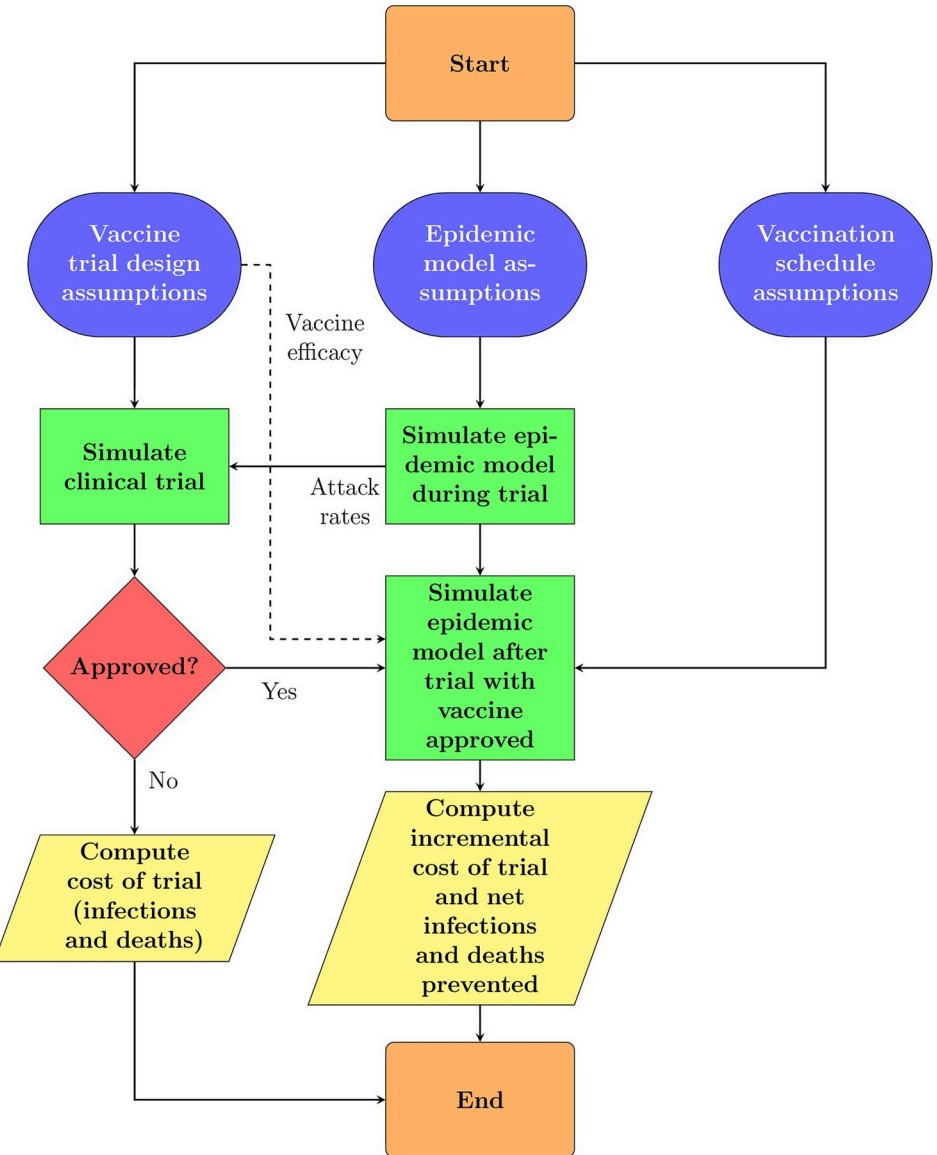

**Fig 1. Simulation framework.** For each Monte Carlo simulation path, we simulate patient-level infections data based on input trial design assumptions and attack rates from the population epidemiological model (for an RCT, ORCT, and ARCT). At the end of the trial (or, at each interim analysis for an ARCT), we determine if the vaccine candidate is approved under superiority or superiority-by-margin testing. Finally, we compute the expected net value of the trial design over 100,000 simulation paths.

vaccine candidate to be some time in November 2021 with a traditional RCT (476 days), between June and August 2021 with an ORCT (326 to 380 days), between April and June 2021 with an ARCT (246 to 306 days), and between March and June 2021 with an HCT (221 to 311 days). For specificity, we report estimated times to licensure using calendar dates and provide the corresponding number of days in parentheses. However, our simulations do depend on calendar dates in one respect: the epidemiological model used to estimate the attack rates depends on current data. Therefore, the estimates reported in this paper are all based on extrapolated conditions as of August 1, 2020, and may need to be revised for other start dates.

The ARCT provides the greatest expected net benefit among the three RCT designs in almost all scenarios. The utility of an HCT versus the RCTs, however, depends critically on the HCT set-up time and the course of the epidemic. The benefits of HCTs are greatest when trials are initiated as early in an epidemic as possible, and/or if the rate of infection is relatively low. Assuming a 30-day set-up time, a vaccine efficacy of 50%, a behavioral epidemiological model, and a population vaccination schedule of 10M doses per day, an HCT can reduce the time to licensure by one month, thus preventing approximately 1.1M incremental infections and 8,000 incremental deaths compared to the best performing alternative clinical trial design, the ARCT. We observe similar results when superiority-by-margin tests are used instead.

We review the designs and assumptions for the four vaccine trials considered in Section 2 and explain our cost/benefit calculations in Section 4. We present the epidemiological model used in our forecasts in Section 2 and report our simulation results in Section 4. We discuss our findings and some broader issues of COVID-19 clinical trials in Section 5 and conclude in Section 6.

## 2 Vaccine trial design

We begin by describing the assumptions and calibrations used in each of the four vaccine trial designs we considered in our simulations.

### 2.1 Traditional vaccine efficacy RCT

First, we consider a traditional double-blind vaccine efficacy trial design. We assume that a closed cohort of 30,000 infection-free but at-risk healthy U.S. adults aged between 18 and 50 years will be enrolled for the study, similar to the phase 3 studies planned or underway for the COVID-19 vaccines developed by Moderna [15], AstraZeneca [16], Pfizer/BioNTech [17], and others. The participants will be randomized equally between the treatment and control arms, receiving either the vaccine candidate or an active control vaccine (e.g., vaccine against meningococcal bacteria), respectively. The use of an active vaccine (e.g., vaccine against meningococcal bacteria) as control provides some benefit to the participants, making it more ethical. It also serves to ensure that the participants are unable to tell whether they received the COVID-19 vaccine based on side effects such as soreness at the injection site, reducing the possibility of behavioral changes that can bias the results of the study. Unlike clinical trials for cancer therapeutics where patient accrual can be a challenge due to the small pool of afflicted patients and strict inclusion/exclusion criteria, subject enrollment for vaccine efficacy studies are often accelerated because there is a large pool of healthy adult volunteers to recruit from. Therefore, we assume an accrual rate of 250 patients per day in our simulations.

Similar to the design of study protocols adopted for phase 3 clinical trials of current leading SARS-CoV-2 vaccine candidates, we assume a hypothetical COVID-19 vaccine candidate that will be administered to subjects in two doses, 28 days apart, i.e., the prime-boost regimen [18, 19]. Furthermore, we assume that it takes approximately 28 days after the booster dose for antibodies to develop (i.e., seroconversion) before surveillance can begin.

We consider efficacy in the prevention of infection by SARS-CoV-2 as the primary endpoint in our study. To draw meaningful conclusions from the trial results, volunteers must be monitored long enough for a sufficient number of infections to occur. Here, we assume a fixed post-vaccination surveillance period of 180 days for all subjects in the cohort, after which a single safety and primary efficacy analysis will be performed to determine licensure (see Section A1 in the S1 Appendix).

Finally, we assume an interval of 120 days after surveillance for the preparation of a biologics license application (BLA) submission to the FDA, including an analysis and publication of

safety, immunogenicity, and efficacy results; collection of chemistry, manufacturing, and controls (CMC) data; the writing of a clinical study report; and subsequent review by the FDA. Under these assumptions, we estimate the time to licensure of our hypothetical candidate under a traditional RCT to be approximately 476 days. This is the baseline value against which we will compare the other three trial designs.

## 2.2 Optimized vaccine efficacy RCT

Depending on the transmission rate of COVID-19 during the trial and the assumed efficacy of the hypothetical candidate, a shorter surveillance period might be sufficient to observe significant results. In general, the higher the transmission rate, the shorter the surveillance period required to observe a statistically significant difference in infection risk between the treatment arm and the control arm (or the lack of thereof) at the same level of significance and power, assuming a constant sample size and vaccine efficacy.

Therefore, we consider an optimized version of the traditional vaccine efficacy RCT design (ORCT) in which the surveillance period is optimized between 30 to 180 days based on different epidemiological scenarios and vaccine efficacies to maximize the expected number of incremental infections and deaths prevented. We note that there is also a trade-off between time and power: A shorter surveillance period will, *ceteris paribus*, reduce the power of the RCT. However, it will also reduce the time to licensure of the vaccine (if approved), which can potentially prevent more infections and save more lives. Conversely, a longer surveillance period will increase the power of the RCT and prolong the time it takes for the vaccine to be approved (see Fig A.4 in the S1 Appendix for an illustration). Apart from the surveillance period, we assume that the ORCT is identical to the RCT in all other aspects.

## 2.3 Adaptive vaccine efficacy RCT

An adaptive version of the traditional vaccine efficacy RCT design (ARCT) is based on group sequential methods [20]. Instead of a fixed study duration with a single final analysis at the end, we allow for early stopping for efficacy via periodic interim analyses of accumulating trial data (see Section A2 in the S1 Appendix). While this reduces the expected duration of the trial, we note that adaptive trials typically require more complex study protocols which can be operationally challenging to implement for test sites unfamiliar with this framework. In our simulations, we assume a maximum of six interim analyses spaced 30 days apart, with the first analysis performed when the first 10,000 subjects have been monitored for at least 30 days. While we have assumed interim analyses at periodic calendar time points here, we note that most vaccine efficacy trials are event based, e.g., performing interim analyses when pre-specified numbers of events occur. In addition, we have adopted Pocock's test for sequential testing (see Section A2 in the S1 Appendix), but we note that some companies are using variants of the O'Brian-Fleming test [21], which have stricter requirements for early stopping, and therefore may lead to longer studies [20].

## 2.4 HCT

Unlike traditional vaccine efficacy field trials which require large sample sizes to observe significant results, we assume that the HCT requires only 250 volunteers, randomized 4:1 between the treatment and control arms. Furthermore, to minimize the risk to participants, we assume that this study will recruit only young and healthy adults aged between 18 and 25 years without any underlying chronic conditions because this group of individuals has the lowest risk of mortality and complications after recovering from the infection [22–24].

It is clear that extensive preparations are required to set up an HCT: selecting, developing, and testing an appropriate challenge virus strain among multiple lineages of SARS-CoV-2; manufacturing a batch of the selected challenge strain under good manufacturing practices (GMP); and identifying the dose level required to achieve satisfactory attack risk of non-severe clinical illness [24] (see Section A12 in the S1 Appendix). From discussions with challenge trial experts, there seems to be a lack of consensus on the appropriate set-up time for HCTs. We reflect this uncertainty in our simulations by incorporating a lag time for HCTs ("set-up time") that ranges between 30 to 120 days.

In the challenge study, volunteers are deliberately exposed to the SARS-CoV-2 virus, reducing post-vaccination monitoring times because investigators do not need to wait for infections to occur naturally as with non-challenge RCTs. Therefore, we assume a surveillance period of only 14 days (the incubation period for COVID-19 [25–27]) for the challenge study. Moreover, the attack rate in the control arm will be independent of the population epidemiological model since the study will be conducted in isolated facilities. In our simulations, we assume that 90% of the subjects in the control arm will be infected after the challenge. We do not assume a 100% attack rate since the challenge strain used is likely weakened to reduce risk to volunteers, and some individuals might have innately stronger immune systems that can counteract the virus.

We note that the FDA is unlikely to approve an experimental vaccine tested in only 200 subjects (versus thousands in non-challenge RCTs), hence we assume that a large-scale safety study will be performed immediately after the conclusion of the challenge study—conditional on positive efficacy results—to evaluate the safety of the hypothetical vaccine candidate in a broader population. Assuming a single-arm study with 5,000 subjects followed for 30 days, we expect the process to be completed in 106 days. To accelerate licensure, we assume that the collection of safety data will be performed in parallel with BLA submission and FDA review. Since the latter is assumed to take 120 days, the additional safety study does not actually add to the time to licensure of the vaccine candidate. It does, however, add to the financial costs of the HCT (see Section A4 in the S1 Appendix).

Apart from the sample size, randomization ratio, set-up time, surveillance period, and safety data requirement, we assume that the HCT is identical to the RCT in all other respects. See Section A3 in the S1 Appendix for a summary of our assumptions.

We anticipate similar post-marketing commitments for both the HCT and the RCTs, in terms of the collection of additional safety and effectiveness data, and supplementary studies to support the effectiveness of the vaccine in populations not included in the initial efficacy study, e.g., infants. However, we do not model them here because they do not affect our cost/benefit computations.

## 3 Epidemiological model

To estimate the attack rate encountered by subjects in a given clinical trial—a key component for our cost/benefit calculations—we require information about the spread of the COVID-19 epidemic in the U.S. We use the Susceptible-Infected-Resolving-Dead-ReCovered with social distancing (SIRDC-SD) model proposed by Fernandez-Villaverde and Jones [28], chosen because it is able to fit both the cumulative and daily number of deaths in all the states well despite being a simple model, to establish a baseline for the epidemic. The details of the model are described in Section A5 in the S1 Appendix.

We estimate the model for each of the 50 states in the U.S. and Washington, D.C., using the time series of deaths in the U.S. obtained from the John Hopkins Center for Systems Science and Engineering (CSSE) COVID-19 repository [29, 30]. Our data were downloaded on June

16, 2020. We do not scale the number of deaths but continue to perform a centered moving average smoothing on the daily number of deaths, as described in Fernandez-Villaverde and Jones [28]. Our estimation method is detailed in Section A6 in the S1 Appendix and the estimated parameters are reported in Table A.3 in S1 Appendix.

The estimated models show how the epidemic has played out thus far but we will need to predict how it will evolve in the future after the lockdowns are relaxed and/or vaccines are developed. To do so, we extend the SIRDC-SD model to take into account semi-effective vaccination. The new model, which we shall name Susceptible-Infected-Resolving-Dead-ReCovered-Vaccinated with social distancing (SIRDCV), is explained in Section A8 in the S1 Appendix.

### 3.1 Evolution of epidemic with reopening

We consider three different scenarios for the evolution of the epidemic over time. In the first, we assume that the current situation will continue indefinitely until the end of the epidemic ("status quo"). That is, stay-home orders and bans on social gatherings will be extended until there are no new infections. We simply forecast ahead of time using the estimated parameters in this scenario.

In the second, we consider that there will be a partial reopening with strict monitoring across all states starting from June 15, 2020 ("ramp"). To model this, we assume a ramp function for $\beta(t)$ that will increase to 0.22 over 90 days and maintain at that level until the end of the epidemic. The parameters are chosen to imply a final $R_0$ of 1.1, which reflects close monitoring and contact tracing, and if needed, temporary quarantines to arrest clusters of infections that may pop up. The contact rate parameter, $\beta$, in this scenario is described by Eq. A.39 in the S1 Appendix.

In the third, we consider the behavioral-based response proposed by John Cochrane ("behavioral"), whereby people voluntarily reduce social contact when they perceive danger (e.g., when they observe that there is an uptick in the daily number of deaths) and increase social contact when they observe that there is a decrease in risk (e.g., when they observe a reduction in the daily number of deaths) [31]. The functional form of $\beta$ is given by Eq A.42 in S1 Appendix.

We give an example of how the basic reproduction number, or $R_0$, may look for each of the scenarios in Fig. A.3 in S1 Appendix.

### 3.2 Population vaccination schedule

We assume that vaccines will be immediately available for distribution and inoculation upon licensure. This reflects how the leading vaccine companies have been scaling up their manufacturing capabilities and started producing millions of doses at industrial scale in parallel to the clinical trials [32, 33] and well before the demonstration of vaccine efficacy and safety. We model three ways that the susceptible population will be vaccinated upon vaccine licensure: 1M, 10M, and infinite doses administered per day. In the last case, the entire U.S. population is assumed to be vaccinated the day after licensure. While unrealistic, this gives an upper bound on the potential benefit of vaccine development. We assume that the vaccines are distributed proportionally to states according to their relative population at the start of the epidemic.

### 3.3 Forecasting infections and deaths

We forecast the cumulative number of confirmed infections and confirmed deaths in each state between February 29, 2020, and December 31, 2022, using the SIRDCV described by

Eq. A.32 to Eq. A.38 in the S1 Appendix before summing over all states in order to produce estimates for the entire U.S. The attack rate at time $t$ is the ratio of the number of new confirmed infections at time $t$ to the number of susceptible persons at time $t − 1$. Illustrations of how the cumulative number of infections and deaths change over time given the different evolution paths of the epidemic and vaccination schedules are shown in Fig. A.2 in S1 Appendix.

## 4 Results

Given the parameters for each trial design and an epidemiological model, we simulate the outcome of hypothetical clinical trials for all four designs and measure their incremental differences.

### 4.1 Cost/Benefit analysis

We apply cost benefit analysis to quantify and compare the net value of each trial design. We focus on public health outcomes—that is, the risks of mortality and morbidity—and provide a qualitative discussion of the societal and financial impact in Section 5.

As shown by Montazerhodjat et al. [34], Isakov et al. [35], and Chaudhuri et al. [36], the value associated with a pathway is computed as the difference between the post-trial benefit and the in-trial cost (Eq 1). The former estimates the net benefit of the trial to society at large while the latter measures the cost of conducting the study to volunteers in the trial.

$$\text{Net Value} = \text{Post-trial Benefit} - \text{In-trial Cost} \tag{1}$$

We quantify the cost of a trial design in terms of the number of COVID-19 infections and deaths observed in the clinical study. For post-trial benefit, we first consider a baseline scenario in which a vaccine is never developed and the epidemic is allowed to run its course. Next, we simulate the case where a vaccine is approved at some point in time depending on the duration of the trial design. The post-trial benefit is then the difference in the cumulative number of infections and deaths in the population between the two scenarios, i.e., the incremental number of infections and deaths prevented with a vaccine licensure.

In our simulations, we consider a vaccine candidate with some true efficacy $\epsilon$ and assume that infections in the clinical study follow a stochastic process (e.g., binomial distribution). Due to this randomness, false rejections of the efficacious vaccine might occur. This is also known as type II error. The false negative rate depends on the trial design (e.g., sample size, surveillance period, maximum type I error, superiority testing) and the epidemiological model (e.g., attack rate in the clinical study). In cases where the vaccine candidate is rejected, net value will be negative since post-trial benefit is zero but cost has been incurred for conducting the clinical trial. Lastly, we assume that the hypothetical vaccine candidate is generally well tolerated and any vaccine-related adverse reactions are mild and negligible with respect to in-trial costs and post-trial benefits [37–39].

### 4.2 Simulation results

We compute the expected net value of different trial designs using Monte Carlo simulations and asymptotic distributions of the efficacy test statistics (see Section A1 in the S1 Appendix). Fig 1 illustrates the inputs, computations, and outputs of our simulation framework. We assume that all trials start on August 1, 2020, and simulate the epidemiological models until December 31, 2022. We perform sensitivity analysis over a wide range of trial design, epidemiological model, and population vaccination schedule assumptions (see Table 1), covering 756 different scenarios. We summarize our results in Table 2 and Section A11 in the S1 Appendix.

**Table 1. Sensitivity analysis with respect to trial design, epidemiological model, and population vaccination schedule assumptions.**

| Parameter | Values | Number of Combinations |
|---|---|---|
| Trial design | RCT, ORCT, ARCT, HCT (30-day set-up), HCT (60-day set-up), HCT (90-day set-up), HCT (120-day set-up) | 7 |
| Vaccine efficacy of hypothetical candidate (%) | 30, 50, 70, 90 | 4 |
| Efficacy requirement | Superiority, superiority by margin of 30% [5], superiority by margin of 50% | 3 |
| Epidemiological scenario | Status quo, ramp, behavioral | 3 |
| Population vaccination schedule (doses/ day) | 1M, 10M, infinite | 3 |

The total number of configurations simulated is 756 (computed as the product of the last column).

In addition to our results, we release an open-source version of our simulation software, and encourage readers to rerun our simulations with their own preferred set of assumptions and inputs.

Assuming superiority testing and a vaccine efficacy of 50%, we estimate the date of licensure of the hypothetical vaccine candidate to be some time in November 2021 under an RCT

**Table 2. Expected number of incremental infections and deaths avoided in the U.S. under different trial designs, vaccine efficacies, and epidemiological scenarios, assuming trials start on August 1, 2020, superiority testing, and 10M doses of a vaccine per day are available after licensure, compared to the baseline case in which no vaccine is ever approved.**

| | Vaccine Efficacy (%) | | | | | | | |
|---|---|---|---|---|---|---|---|---|
| | 30 | | 50 | | 70 | | 90 | |
| | $\mathbb{E}[\Delta\text{Infections}]$ | $\mathbb{E}[\Delta\text{Deaths}]$ | $\mathbb{E}[\Delta\text{Infections}]$ | $\mathbb{E}[\Delta\text{Deaths}]$ | $\mathbb{E}[\Delta\text{Infections}]$ | $\mathbb{E}[\Delta\text{Deaths}]$ | $\mathbb{E}[\Delta\text{Infections}]$ | $\mathbb{E}[\Delta\text{Deaths}]$ |
| **Status Quo** | | | | | | | | |
| RCT | 3,914 | 31 | 11,539 | 92 | 19,130 | 151 | 21,557 | 170 |
| ORCT | 5,589 | 45 | 16,802 | 134 | 33,757 | 269 | 50,288 | 401 |
| ARCT | 9,596 | 76 | 31,473 | 250 | 66,641 | 531 | 83,522 | 665 |
| HCT (30-day set-up) | 140,731 | 1,124 | 152,263 | 1,216 | 156,885 | 1,254 | 159,876 | 1,277 |
| HCT (60-day set-up) | 110,046 | 879 | 118,937 | 950 | 122,482 | 979 | 124,777 | 997 |
| HCT (90-day set-up) | 86,466 | 690 | 93,370 | 745 | 96,111 | 768 | 97,886 | 782 |
| HCT (120-day set-up) | 68,213 | 544 | 73,611 | 587 | 75,747 | 605 | 77,132 | 615 |
| **Behavioral** | | | | | | | | |
| RCT | 363,382 | 2,845 | 386,081 | 3,026 | 397,396 | 3,117 | 404,562 | 3,174 |
| ORCT | 1,139,585 | 9,061 | 1,377,157 | 10,955 | 1,426,014 | 11,345 | 1,457,500 | 11,598 |
| ARCT | 2,588,881 | 20,647 | 3,248,449 | 25,924 | 3,389,541 | 27,052 | 3,473,035 | 27,720 |
| HCT (30-day set-up) | 3,903,566 | 31,167 | 4,309,316 | 34,411 | 4,481,448 | 35,789 | 4,591,750 | 36,671 |
| HCT (60-day set-up) | 2,795,316 | 22,301 | 3,082,676 | 24,598 | 3,205,159 | 25,579 | 3,283,975 | 26,209 |
| HCT (90-day set-up) | 2,011,244 | 16,028 | 2,211,985 | 17,633 | 2,297,350 | 18,316 | 2,352,436 | 18,757 |
| HCT (120-day set-up) | 1,466,239 | 11,668 | 1,605,833 | 12,784 | 1,664,613 | 13,255 | 1,702,601 | 13,558 |
| **Ramp** | | | | | | | | |
| RCT | 1,075,634 | 8,316 | 1,131,531 | 8,764 | 1,160,564 | 8,996 | 1,179,234 | 9,145 |
| ORCT | 2,853,202 | 22,569 | 3,839,945 | 30,432 | 3,973,769 | 31,501 | 4,050,013 | 32,111 |
| ARCT | 5,711,310 | 45,401 | 7,442,922 | 59,253 | 7,924,650 | 63,107 | 8,071,866 | 64,285 |
| HCT (30-day set-up) | 8,744,377 | 69,672 | 9,452,413 | 75,330 | 9,725,022 | 77,511 | 9,897,591 | 78,892 |
| HCT (60-day set-up) | 6,814,762 | 54,235 | 7,381,425 | 58,762 | 7,602,878 | 60,534 | 7,743,514 | 61,659 |
| HCT (90-day set-up) | 5,266,925 | 41,851 | 5,711,663 | 45,404 | 5,887,421 | 46,811 | 5,999,381 | 47,706 |
| HCT (120-day set-up) | 4,053,134 | 32,141 | 4,396,033 | 34,879 | 4,532,400 | 35,970 | 4,619,521 | 36,667 |

(476 days), between June and August 2021 under an ORCT (326 to 380 days), between April and June 2021 under an ARCT (246 to 306 days), and between March and June 2021 under an HCT (221 to 311 days). Apart from an RCT which has a fixed trial duration, the dates of licensure from the ORCT and ARCT depend largely on the status of the epidemic during the clinical trial. If the transmission rate of the disease is low (e.g., due to social distancing or other non-pharmaceutical interventions), an extended surveillance period is required to accrue enough natural infections in order to observe a statistically significant difference in infection risk between the treatment arm and the control arm. Conversely, when the transmission rate is high, a short surveillance period is sufficient to observe significant results. We note that an HCT, on the other hand, does not depend on the epidemic situation but is instead limited by the time required to set up the challenge model. In general, we find that the time to licensure under ORCT and ARCT decreases with increasing vaccine efficacy: the greater the efficacy, the easier it is to observe a significant treatment effect.

We find that the ARCT provides the greatest expected net benefit among the three RCT designs in almost all scenarios. The utility of an HCT versus the RCTs, however, depends critically on the set-up time and the dynamics of the epidemic. For example, assuming superiority testing, a vaccine efficacy of 50%, the behavioral epidemiological model, and a population vaccination schedule of 10M doses per day, we estimate that the ARCT can help accelerate licensure by almost 8 months versus the RCT, thus preventing approximately 2.9M incremental infections and 23,000 incremental deaths from COVID-19 in the U.S. versus the latter.

Under the same set of assumptions, an HCT that requires 30 days to set up can *further* reduce the time to licensure by a month, thus preventing approximately 1.1M more infections and 8,000 more deaths versus the ARCT. However, the advantage of the HCT vanishes when its set-up time is long: an HCT that requires 90 days to set up takes about one month longer to reach licensure as compared to the ARCT, leading to around 1.0M more infections and 8,000 more deaths versus the latter (see Fig 2a). Under such circumstances, the use of an HCT is worthwhile only when the prevalent transmission rate is low. If we consider the status quo scenario instead of the behavioral epidemiological model, the time to licensure is about one month shorter under the HCT than under the ARCT even with a 90 day set-up period (see Fig 2b). In this case, the HCT prevents approximately 60,000 incremental infections and 500 incremental deaths versus the ARCT. We observe similar trends under superiority-by-margin testing at a threshold of 50%.

## 5 Discussion

There has been a plethora of papers highlighting various ethical considerations for conducting HCTs [40, 41], some specifically for COVID-19 [9, 42–46]. Some of the main ethical concerns are: (1) what is the explicit scientific rationale for, and societal value of, an HCT; (2) whether the risks of harm to the subjects and the public at large are understood by the scientists and have been minimized; (3) whether informed consents have been obtained from subjects after they are given full disclosures of the risks involved; and (4) whether the subjects have been selected fairly and given appropriate compensation for both the risk and actual harm brought on by HCTs. Most bioethicists generally accept that these concerns can be addressed within the existing ethical framework for human medical research.

Our paper addresses the first and second of these ethical concerns. We provide scientific justifications for COVID-19 HCTs by considering how conducting them can allow companies to learn about the protection curves and accelerate the development of vaccines against SARS-CoV-2.

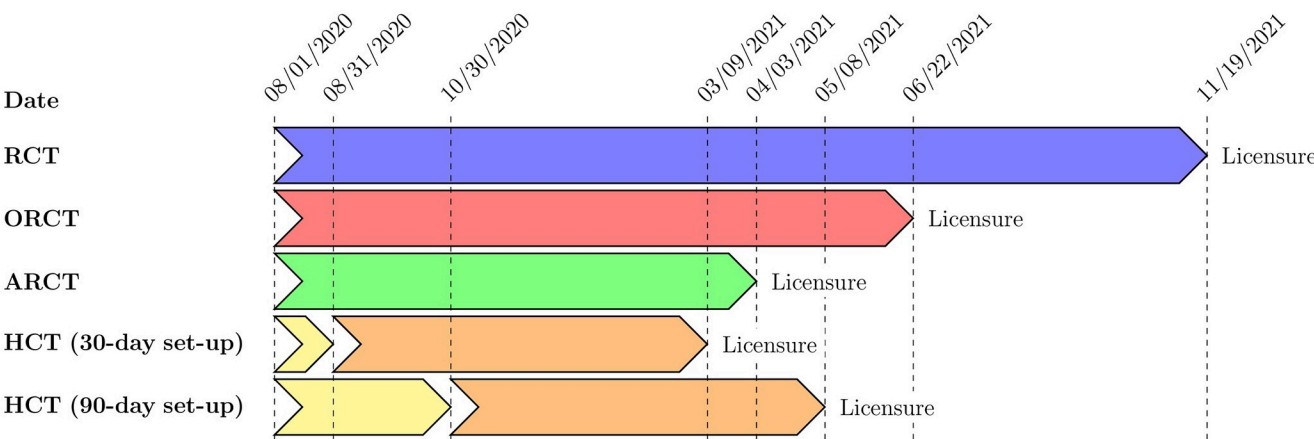

(a) Under the behavioral epidemiological model.

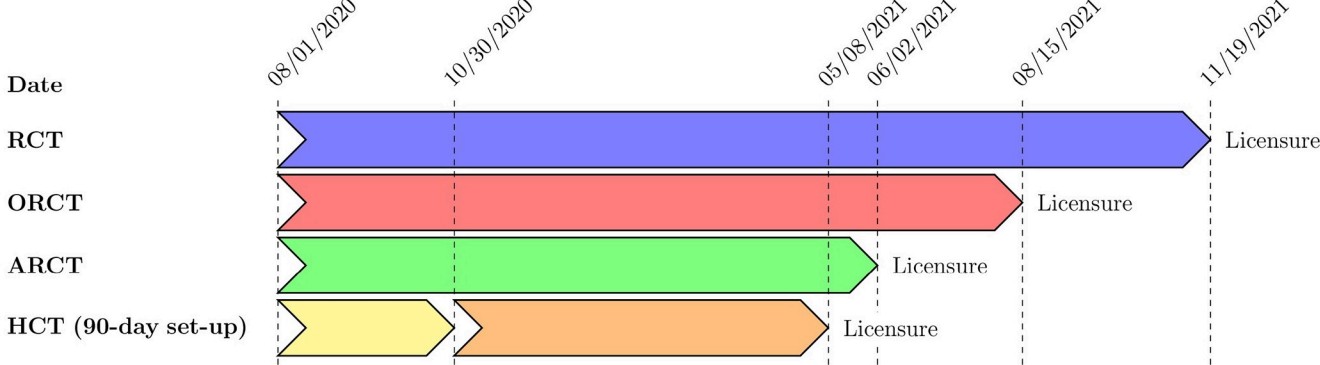

(b) Under the status quo epidemiological model.

**Fig 2. Dates of licensure under RCT, ORCT, ARCT, HCT (30-day set-up time), and HCT (90-day set-up time), assuming superiority testing, a vaccine efficacy of 50%, and a population vaccination schedule of 10M doses per day.** (a) Under the behavioral epidemiological model. (b) Under the status quo epidemiological model.

However, our analysis does not address the latter two ethical considerations as they concern the execution of HCTs, which is beyond the scope of this paper. Nonetheless, companies and scientists seeking to perform HCTs, and especially regulators, will have to address those concerns to preserve public trust and avoid a public backlash that could jeopardize other important medical research critical to addressing the current epidemic.

Some scientists argue that "a single death or severe illness in an otherwise healthy volunteer would be unconscionable" [46]. However, it can be argued that allowing tens of thousands of individuals to die by denying the consent of an informed individual to take a calculated risk is equally unconscionable. In this paper, we adopt the Benthamite approach [47], where every individual's utility is weighted equally in the aggregate utility function, as is the common convention in public economics analyses. Within this ethical perspective, our calculations show that an HCT can potentially provide substantial public health benefits in terms of accelerating vaccine development and reducing the burden of coronavirus-related mortality and morbidity in the U.S.—in some cases, by more than 1.1M infections and 8,000 deaths compared to the best performing RCT—when conducted early in the pandemic's life

cycle and in cases where the spread of COVID-19 in the population is muted due to non-pharmaceutical interventions.

We also expect the financial costs of an HCT—which includes the cost of liability protection—to be lower than those of a traditional vaccine efficacy RCT, adding further support for a challenge design (see Section A4 in the S1 Appendix for further discussion). While we have focused on public health outcomes here, it is clear that accelerated vaccine development provides tremendous societal and economic benefits as well—e.g., savings in insured medical costs, direct medical expenditures, and hospitalization costs, and accelerated economic recovery from an earlier reopening.

We emphasize that the expected costs and benefits of a clinical trial depend critically on many assumptions about existing conditions. For example, recruiting subjects in sufficient numbers and diversity can sometimes present a challenge for clinical trials involving experimental vaccines (although, in the case of HCTs for COVID-19, the organization 1Day Sooner reports over 32,000 registered volunteers as of July 27, 2020). Also, we do not include set-up time for non-challenge RCTs because phase 3 vaccine efficacy trials are already imminent as of now. Moreover, we assume a relatively short set-up time for HCTs because a challenge study can be set up relatively quickly using a wild-type strain [24], and the National Institute of Allergy and Infectious Diseases (NIAID) appears to have already made some headway in manufacturing challenge doses [48–50]. If, instead, we assume comparable set-up times (e.g., two months) and start dates for both an HCT and non-challenge RCTs, we expect that an HCT can accelerate licensure by two months when compared to an adaptive RCT, assuming superiority testing, a vaccine efficacy of 50%, and the behavioral epidemiological model. Some have argued that at least one to two years is required to develop a robust model from scratch [46]. In this case, our results indicate that an ARCT will almost always be faster than an HCT. However, even if an HCT with a long set-up time does not lead to faster vaccine licensures over an ARCT given current conditions, the creation of a standing HCT agent and setting up an HCT now can provide a hedge against potential failures in the current crop of vaccine candidates. By having an approved, ready-to-go challenge virus and ready-to-go HCT sites that vaccine developers can access immediately, the approval process for as-yet-untested SARS-CoV-2 vaccine candidates can be accelerated when required. For a pandemic like COVID-19, such a hedge will almost always show substantial net benefits relative to its costs.

HCTs have several other benefits that will be more obvious as the pandemic progresses. They require many fewer eligible volunteers, whose numbers will dwindle as the pandemic progresses. They do not depend on attack rates at clinical trial sites which are notoriously difficult to estimate and highly dependent on non-pharmaceutical interventions such as lockdowns and other social-distancing policies. They also avoid logistical problems such as identifying subjects, obtaining subjects' consent, obtaining institutional review board's approval or tracking subjects, particularly when attempting large-scale clinical trials in places where contract research organizations (CROs) have little experience.

It is conceivable that multiple vaccines—instead of the single vaccine in our simulation study—are tested concurrently in a single trial design [51]. For example, five vaccines, such as those selected by Operation Warp Speed [52], could be tested concurrently in a six-arm trial (five vaccine arms and a control arm), requiring 40% fewer test subjects, thereby reducing in-trial expected morbidity and mortality costs by the same amount. The benefits can be increased if an adaptive platform clinical trial—designed to eliminate ineffective vaccines at the first signs of futility—is adopted. A clinical trial testing multiple vaccines can also reduce competition for volunteers, a problem that continues to plague vaccine developers [53].

We choose to quantify the cost and benefits of the clinical trials by measuring the number of infections and deaths avoided, and refrain from performing a traditional health technology

assessment, such as comparing the economic value of an HCT versus an RCT using quality-adjusted life years measures or willingness to pay estimates such as the value of a statistical life. Performing such computations is straightforward given the output of our simulations, but we have refrained from doing so in deference to non-economist stakeholders who find it offensive to use any pecuniary measures when discussing the loss of human life.

Finally, our analysis focuses mainly on the U.S. for practical reasons involving access to data with which to calibrate our simulations and the broader goal of informing U.S. public health officials and policymakers as the country enters the final stages of vaccine development. However, we note that vaccine companies such as Pfizer/BioNTech are also looking at recruitment in the Southern Hemisphere (e.g., Brazil [54]), which can affect the rate at which these events accumulate in trials, depending on the spread of COVID-19 in those countries. In addition, a vaccine licensure may apply internationally. Given that the U.S. currently comprises 25% of all confirmed COVID-19 cases (as of July 7, 2020) [29], if the assumptions made in our study also hold internationally, the net benefits for all the clinical trials will scale by a factor of 4, in which case HCTs can save an additional 4.4M infections and 32,000 deaths compared to the best performing RCT in certain situations.

We highlight that these figures depend heavily on the development of the epidemic in the U.S. moving forward. We have considered three simple scenarios, status quo, ramp, and behavioral, corresponding to low transmission, moderate transmission, and behavioral-based response, respectively. There are clearly many other sources of uncertainty that are not reflected here. For example, non-adherence to social distancing advisories and/or resistance to precaution recommendations such as wearing a mask in public will lead to an uncontrolled outbreak, which will help to accelerate non-challenge RCTs, making them attractive even when compared to an HCT with a short set-up time. We have found it difficult and impractical to incorporate these uncertainties in our assumptions due to the speed at which things are evolving and the unpredictability of public reaction. In addition, studies that have attempted to incorporate such uncertainties in their epidemic model report huge error bounds in their projections [55]. The wide confidence intervals prevent us from drawing any useful conclusions, which severely limit the usefulness of such models. Therefore, we recommend readers not to take our results as final or definitive, but to re-run our simulations with their own preferred set of assumptions, calibrated using the most current epidemiological data.

## 6 Conclusion

Our paper presents a systematic framework for quantitatively accessing the in-trial and societal cost/benefit trade-offs of various clinical trial designs in terms of infections and deaths averted. We hope that this framework will allow stakeholders to make more informed practical and ethical decisions regarding accelerating COVID-19 vaccine development in the ongoing pandemic.

## Supporting information

**S1 Appendix. In the appendix, we include detailed results about clinical trial design, epidemiological models, and additional simulation results.**
(PDF)

## Acknowledgments

We thank Arthur Caplan, Norman Baylor, and Frederick Hayden for helpful comments and discussion, Amanda Hu for research assistance, and Jayna Cummings for editorial support.

The views and opinions expressed in this article are those of the authors only and do not necessarily represent the views and opinions of any other organizations, any of their affiliates or employees, or any of the individuals acknowledged above.

## Author Contributions

**Conceptualization:** Andrew W. Lo.

**Data curation:** Kien Wei Siah, Chi Heem Wong.

**Formal analysis:** Kien Wei Siah, Chi Heem Wong.

**Investigation:** Kien Wei Siah, Chi Heem Wong.

**Methodology:** Donald A. Berry, Scott Berry, Peter Hale, Leah Isakov, Andrew W. Lo, Kien Wei Siah, Chi Heem Wong.

**Resources:** Andrew W. Lo.

**Software:** Kien Wei Siah, Chi Heem Wong.

**Supervision:** Andrew W. Lo.

**Visualization:** Kien Wei Siah, Chi Heem Wong.

**Writing – original draft:** Kien Wei Siah, Chi Heem Wong.

**Writing – review & editing:** Donald A. Berry, Scott Berry, Peter Hale, Leah Isakov, Andrew W. Lo, Kien Wei Siah, Chi Heem Wong.

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
