## [Decision Letter · Decision Letter 0]

26 Oct 2020

PONE-D-20-25637

A cost/benefit analysis of clinical trial designs for COVID-19 vaccine candidates

PLOS ONE

Dear Dr. Lo,

Thank you for submitting your manuscript to PLOS ONE. After careful consideration, we feel that it has merit but does not fully meet PLOS ONE’s publication criteria as it currently stands. Therefore, we invite you to submit a revised version of the manuscript that addresses the points raised during the review process.

We look forward to receiving your revised manuscript.

Kind regards,

Xia Jin, MD, PhD

Academic Editor

PLOS ONE

Journal Requirements:

"We thank Arthur Caplan for helpful comments and discussion, Amanda Hu for research 493

assistance, and Jayna Cummings for editorial support. The views and opinions 494

expressed in this article are those of the authors only and do not necessarily represent 495

the views and opinions of any other organizations, any of their affiliates or employees, or 496

any of the individuals acknowledged above. Funding support from the MIT Laboratory 497

for Financial Engineering is gratefully acknowledged, but no direct funding was received 498

for this study and no funding bodies had any role in study design, data collection and 499

analysis, decision to publish, or preparation of this manuscript. The authors were 500

personally salaried by their institutions during the period of writing (though no specific 501

salary was set aside or given for the writing of this manuscript). 502

Conflict of Interest Disclosure

P.H., K.S., and C.W. report no conflicts.

L.I. is an employee of the biotech company Seqirus and receives salary and company

stock as part of compensation

A.L. reports personal investments in private biotech companies, biotech venture capital

funds, and mutual funds. A.L. is a co-founder and partner of QLS Advisors, a healthcare

analytics and consulting company; an advisor to BrightEdge Ventures; a director of

BridgeBio Pharma, Roivant Sciences, and Annual Reviews; chairman emeritus and

senior advisor to AlphaSimplex Group; and a member of the Board of Overseers at Beth

Israel Deaconess Medical Center and the NIH's National Center for Advancing

Translational Sciences Advisory Council and Cures Acceleration Network Review Board.

During the most recent six-year period, A.L. has received speaking/consulting fees,

August 15, 2020 14/18

honoraria, or other forms of compensation from: AIG, AlphaSimplex Group, BIS,

BridgeBio Pharma, Citigroup, Chicago Mercantile Exchange, Financial Times, FONDS

Professionell, Harvard University, IMF, National Bank of Belgium, Q Group, Roivant

Sciences, Scotia Bank, State Street Bank, University of Chicago, and Yale University.".

i) We note that you have provided funding information that is not currently declared in your Funding Statement. However, funding information should not appear in the Acknowledgments section or other areas of your manuscript. We will only publish funding information present in the Funding Statement section of the online submission form.

ii) Please remove any funding-related text from the manuscript and let us know how you would like to update your Funding Statement. Currently, your Funding Statement reads as follows:

iii) Additionally, because some of your funding information pertains to [commercial funding//patents], we ask you to provide an updated Competing Interests statement, declaring all sources of commercial funding. 

iv) In your Competing Interests statement, please confirm that your commercial funding does not alter your adherence to PLOS ONE Editorial policies and criteria by including the following statement: "This does not alter our adherence to PLOS ONE policies on sharing data and materials.” as detailed online in our guide for authors  http://journals.plos.org/plosone/s/competing-interests.  If this statement is not true and your adherence to PLOS policies on sharing data and materials is altered, please explain how.

 * Please include the updated Competing Interests Statement and Funding Statement in your cover letter. We will change the online submission form on your behalf.

Reviewers' comments:

Reviewer's Responses to Questions

**Comments to the Author**

1. Is the manuscript technically sound, and do the data support the conclusions?

Reviewer #1: Partly

Reviewer #2: Yes

2. Has the statistical analysis been performed appropriately and rigorously? 

Reviewer #1: Yes

Reviewer #2: Yes

3. Have the authors made all data underlying the findings in their manuscript fully available?

Reviewer #1: Yes

Reviewer #2: Yes

4. Is the manuscript presented in an intelligible fashion and written in standard English?

Reviewer #1: Yes

Reviewer #2: Yes

5. Review Comments to the Author

Reviewer #1: This is a timely manuscript that can provide useful information to those debating the ethics of human challenge trials for SARS-CoV-2.

Minor:

Line 222: data were (and check elsewhere)

Substantive:

Lines 2-3: the pandemic has not upended lives of billions, and caused those losses; it is our response, and sometimes lack of response, that did so.

Line 3: replace “unlikely” which is a probabilistic statement with “may” or something similar. And, elsewhere in the manuscript…

Line 68: A true efficacy of 50% is assumed. Under this assumption, no matter how many people are enrolled, there will only be 50% power to meet current FDA guidelines, which include the POINT ESTIMATE being at least 50%.

Lines 87-88—why are superiority-by-30% simulations not the main ones, since they correspond to the FDA guidelines?

Line 99: BioNTech is, and others are or will, recruiting in the Southern Hemisphere as well (e.g. Brazil, Argentina). OK to stick with the US for this comparative exercise, but something could be said in the Discussion.

Lines 124-126, 147-148—The main current trials are event-driven—although they will follow up all participants for a year or so, when there are 150-200 primary events, they will submit for approval. I suppose that with many interim analyses, the results of the simulations would be similar whether based on time or events, but still, with the vagaries of the current pandemic trajectory, nobody would base them on elapsed time.

Optimized section, line 135 on: It is not at all clear what is the design of this study. How is it optimized—how is the decision made as to the length of the study? I have never seen a design described like this, and I did not see any references here to work that has described or used such a design.

Adaptive section, from line 154: I don’t think of this as adaptive, and I don’t think others will either—it is just a standard group sequential design. There is no sample size re-estimation, no changes in arms or doses or recruitment, etc. This is in fact the design the main studies are using, EXCEPT instead of time-based analyses, they are based on numbers of accrued cases, e.g. 40, 80, 120, 160 cases.

Line 181: What does the HCT setup time include? First, there needs to be some kind of dose-ranging study to find out the optimal standardized infective dose—people are already working on this now, in case we do get to HCTs, but I don’t know where they are at or how long it takes.

Epi model section line 211: what is being modeled? Throughout the manuscript, “infections” is used—symptomatic infections, which is the primary endpoint being used by current vaccine studies?

Table 1: what exactly does “superiority by margin of 50%” mean? FDA has specified that the confidence interval should lie above 30%.

Eq A16—shouldn’t those lambdas be lower case?

P. 4 Appendix—use of Pocock’s—some companies are using this, others versions of O’Brien-Fleming, which may lengthen the trial—should at least say something in the Discussion about this.

Although 504 configurations are referred to, I did not see it spelled out anywhere just what these are. Don’t make the reader work to figure out stuff like this. I couldn’t even tell how much the eventual attack rates varied, which is critical to the timecourse. Baseline numbers are from all 50 states, but recruitment is highly varied geographically, and some trials are trying to recruit high-risk individuals—need to know how much these rates are varying so we can see the effect of eventual attack patterns on timelines.

Unfortunately, a great deal of effort has gone into the extensive simulations. We could have had the same results without any simulations—just tabulate a wide range of approximate final attack rates, and show when how many cases would be accumulated under each, and then do the cost-benefit analyses on those. Other groups have made much more extensive simulations of the pandemic; the focus of this manuscript is not coming up with accurate trajectories.

Reviewer #2: This manuscript is well written and informs the statistical rationale for evaluating the common three clinical trial designs against less common (HCT) to be considered for phase III COVD-19 clinical vaccine trials. The designs described are the traditional vaccine efficacy randomized clinical trial (RCT), optimized vaccine efficacy RCT (ORCT) where surveillance period is optimized, adaptive vaccine efficacy RCT (ARCT) with sequential analysis of data, and human challenge trial (HCT) with advantages of smaller participant size and less monitoring time but with more health risk and more set up time. These designs are clearly characterized with minimal biases. However, the authors did not integrate the published information already available before their submission on the four COVID-19 vaccine trials which are ongoing world-wide and supported and/or monitored by NIH. These are the vaccine trials by Moderna, AstraZeneka, Pfizer/BioNTech, and possibly Johnson and Johnson’s Janssen Pharmaceutical. However, the manuscript cites the website announcements of Moderna, AstraZeneka, and Pfizer/BioNTech.

Instead of only citing the company’s website announcements, the authors should have included in their manuscript Moderna’s preliminary report on their phase 1 dose-escalation, open-label trial in New England Journals of Medicine (published on July 14, 2020; DOI: 10.1056/NEJMoa2022483) and AstroZeneka’s phase 1/2 single-blind RCT in Lancet (published online on July 20, 2020; https://doi.org/10.1016/ S0140-6736(20)31604-4). If the authors have read these articles, they would have realized the level of reactogenicity or adverse effects that these two vaccines have in their phase I trials. The AstroZeneka’s vaccine, compared to the licensed meningococcal vaccine control group, had more local and systemic adverse reactions which lasted up to seven days. Both if these phase I trials used healthy 18-55 years old but the severity of such adverse effects is unknown to those who have comorbidity and to older population at the time of manuscript submission. The authors should describe the potential consequences of high adverse vaccine effect. It appears that none of the formula included in the appendix integrates such a risk factor. The risk factor that either the vaccine delivery system used (ChAdOx1 vector, mRNA-LNP, Ad26 vector) and/or the vaccine immunogen expression may have caused higher adverse effect which may also impact the accrual of the participants. More importantly, authors should describe what kind of immunity must be generated by the COVID vaccine to be effective in humans since vaccine efficacy should also be based on immune correlate of vaccine protection.

The phase III trials are all in place for the four companies at the time of manuscript review but should not be required for this manuscript since such information although on Clinicaltrials.gov were confirmed by the companies based on the results from phase I or 1/2 trials which were after the manuscript submission. Nevertheless, none of the four phase II trials uses HCT. In addition, FDA and NIH have not declared in peer-reviewed journal publication(s) that HCT is currently being considered for COVID vaccine. Please do not use citation such as in reference 44 (politico from White House). Please use peer reviewed journal citation for such a controversial subject. In light of not knowing whether CoV-2 spike protein can be safely administered as vaccine immunogen without long-term adverse incident, the use of HCT is a concern. Until safety of the volunteers can be fully met with effective therapy against COVID-19, HCT may not be a consideration for prophylactic COVID-19 vaccine trial. An example of another emerging pathogen that have undergone extensive clinical vaccine trials is the prophylactic HIV-1 vaccines, but HCT has yet to be used. Instead, newer HIV-1 vaccine designs are tested in HIV-1 positive individuals as therapeutic vaccine, which is a more scientifically and ethically rationale approach. The transmission rate of the SARS-CoV-2 is rapid without mask and appropriate distancing. Hence consideration should be in how you recruit the volunteers such as those who wear mask versus those who do not wear mask. Overall, the authors are raising an interesting question of the potential of using HCT based on how much lives saved and level of morbidity decreased which is an interesting subject to be raised to the scientific community without biases but with care.

6. PLOS authors have the option to publish the peer review history of their article (what does this mean?). If published, this will include your full peer review and any attached files.

Reviewer #1: No

Reviewer #2: No

---

## [Author Response · Author response to Decision Letter 0]

25 Nov 2020

Response to Reviewers

We thank the reviewers for providing us with extremely useful and constructive feedback, all of which we have incorporated in our revision, which we believe has greatly improved the manuscript. Below is a point-by-point response to all of the reviewers’ comments (which we have included in boldface for convenience).

Reviewer 1

This is a timely manuscript that can provide useful information to those debating the ethics of human challenge trials for SARS-CoV-2.

1. Line 222: data were (and check elsewhere)

Response:

Thank you for noting this typographical error. We have corrected this throughout the manuscript.

2. Lines 2-3: the pandemic has not upended lives of billions, and caused those losses; it is our response, and sometimes lack of response, that did so.

Response:

Thank you for pointing this out. We have now revised the description on page 1 to be more accurate.

3. Line 3: replace “unlikely” which is a probabilistic statement with “may” or something similar. And, elsewhere in the manuscript…

Response:

Thank you for the suggestion. We have rephrased line 3–4 on page 1 to avoid the use of the word “unlikely.” We have also reviewed its use in other parts of the manuscript but decided to keep them because we feel that the word best describes the points we are trying to make.

4. Line 68: A true efficacy of 50% is assumed. Under this assumption, no matter how many people are enrolled, there will only be 50% power to meet current FDA guidelines, which include the POINT ESTIMATE being at least 50%.

Response:

The power of the clinical trial is related to but not completed determined by the efficacy of the vaccine candidate. It is also dependent on other factors, such as the level of significance, the number of volunteers recruited, the prevalent attack rate, and the length of the surveillance period. For example, a clinical trial testing a vaccine candidate with high true efficacy can still have low power if poorly sized (i.e., small sample size). Similarly, the power of the clinical trial will be low if the surveillance period is too short and/or the transmission rate of COVID-19 is too low to observe a sufficient number of events for a statistically significant result.

5. Lines 87-88—why are superiority-by-30% simulations not the main ones, since they correspond to the FDA guidelines?

Response:

Thank you for pointing this out. We have now updated the appendix to include results for superiority-by-margin of 30% in Appendix A11. We decided to focus on superiority testing in the main manuscript because we believe that such a criterion might be appropriate for emergency use authorization during a pandemic where no alternatives are available. However, for readers who are interested, we have also included results for superiority-by-margin of 30% and 50% in Appendix A11. That being said, we would be happy to change the focus to superiority-by-30% if the reviewer strongly disagrees.

6. Line 99: BioNTech is, and others are or will, recruiting in the Southern Hemisphere as well (e.g. Brazil, Argentina). OK to stick with the US for this comparative exercise, but something could be said in the Discussion.

Response:

Thank you for the suggestion. We have now added a note in the Discussion on page 12 to highlight that BioNTech and other vaccine companies are also looking at recruitment in the Southern Hemisphere such as Brazil and Argentina.

7. Lines 124-126, 147-148—The main current trials are event-driven—although they will follow up all participants for a year or so, when there are 150-200 primary events, they will submit for approval. I suppose that with many interim analyses, the results of the simulations would be similar whether based on time or events, but still, with the vagaries of the current pandemic trajectory, nobody would base them on elapsed time.

Response:

This point is well-taken. For example, the Pfizer/BioNTech study targets a total 164 primary events, with four interim analyses (IAs) to be performed after accrual of 32, 62, 92, and 120 cases.1 The Moderna study targets a total 151 primary events, with two IAs planned after accrual of 53 and 106 cases. They expect the first and second IAs to be performed in December and February, respectively, based on their estimate of the COVID-19 incidence rate.2 Unfortunately, we have found that this approach is less feasible in our framework because we consider not just one set of assumptions (as in what has been done in the study protocols of the Pfizer/BioNTech and Moderna trials), but sweep a wide range of target vaccine efficacies and epidemiological scenarios. Therefore, it is difficult to identify specific event accrual milestones that are valid for all scenarios in our simulations. As an alternative, we consider a design with IAs performed at periodic calendar time points.3 We note that similar designs have been applied in clinical trials testing interventions for COVID-19.4 As the reviewer points out, we believe that with large sample sizes and under multiple IAs, the results for both approaches should be very similar. On page 5 of the manuscript, we also explicitly acknowledge this difference.

8. Optimized section, line 135 on: It is not at all clear what is the design of this study. How is it optimized—how is the decision made as to the length of the study? I have never seen a design described like this, and I did not see any references here to work that has described or used such a design.

Response:

The ORCT is identical to the traditional vaccine efficacy RCT in all aspects except the length of the surveillance period. (See Appendix A3.) In our simulations, we assumed a surveillance period of 180 days for the traditional RCT. However, we note that depending on the transmission rate of COVID-19 during the trial and the efficacy of the vaccine candidate, a shorter surveillance period might be sufficient to observe significant results. For example, the higher the transmission rate, the quicker events accumulate, and therefore, the shorter the surveillance period required to achieve the same level of power.

There is also a trade-off between surveillance time and power. A shorter surveillance period will, ceteris paribus, reduce the power of the clinical trial. With lower power (i.e., lower probability of approval), the expected number of infections and deaths that can be prevented by an efficacious vaccine candidate will be lower due to a larger Type II error. However, a shorter surveillance period also means a shorter time to licensure (in cases where the candidate is approved), which can potentially prevent more infections and save more lives because a vaccine can be made available sooner. Conversely, a longer surveillance period will increase the power of the clinical trial and the expected number of infections and deaths that can be prevented. However, such a trial will only be completed later into the pandemic.

Bearing this in mind, we propose a novel design where the surveillance period is optimized to achieve the maximum net value in terms of the expected infections prevented and lives saved. We perform the optimization for each set of vaccine efficacy and epidemiological scenario assumptions considered in our analysis. (See Appendix A10 for an illustration of the interaction between the length of the surveillance period, power of the clinical trial, and infections avoided.)

This design represents the upper bound of the net benefits that can possibly be achieved by a fixed-duration clinical trial with a single final efficacy analysis and no interim analyses. It is conceptually similar to the framework proposed in Chaudhuri et al. (2020)5 where the Type I errors and sample sizes of clinical trials for anti-infective therapeutics are optimized to minimize the expected loss to patients and the population during an epidemic outbreak.

9. Adaptive section, from line 154: I don’t think of this as adaptive, and I don’t think others will either—it is just a standard group sequential design. There is no sample size re-estimation, no changes in arms or doses or recruitment, etc. This is in fact the design the main studies are using, EXCEPT instead of time-based analyses, they are based on numbers of accrued cases, e.g. 40, 80, 120, 160 cases.

Response:

Thank you for pointing this out. We explicitly acknowledge that the ARCT design is based on group sequential methods on page 4 of the manuscript. We understand that the reviewer does not consider such a design as adaptive. However, in this paper, we define an adaptive design as a clinical trial design that incorporates prespecified opportunities to use accumulating trial data to modify aspects of an ongoing trial, such as early stopping for success at interim analyses using group sequential methods. We have found similar definitions in literature.6

We attempted to incorporate additional adaptive features (e.g., sample size re-estimation and adaptive randomization) but found them largely impractical due to the nature of vaccine efficacy clinical trials. For example, to have adaptive randomization, we would need to consider partial enrollment in several batches over time. Assuming 28 days for vaccination (prime-boost regimen), 28 days for subjects to develop immune response post-vaccination, and 14 days for surveillance (incubation period of COVID-19), the batches must be spaced at least 70 days apart so that results from the first/earlier batch are available to inform the randomization ratio of the next batch. This will prevent the trial from being completed in a timely fashion, which is especially important in a pandemic where there is an urgent need for vaccines and treatments, and where the time-window for conducting large scale clinical trials is limited due to ever-changing attack rates.

10. Line 181: What does the HCT setup time include? First, there needs to be some kind of dose-ranging study to find out the optimal standardized infective dose—people are already working on this now, in case we do get to HCTs, but I don’t know where they are at or how long it takes.

Response:

Extensive preparations are required to set up an HCT. For example, steps include the selection of a SARS-CoV-2 challenge strain, purification and full characterization of the challenge strain, cGMP (current Good Manufacturing Practice) production of the challenge pool, and, as the reviewer mentioned, a dose-ranging study to determine the appropriate infectious dose. We have consulted with several experts who have extensive experience in influenza HCTs to come up with a more detailed list of steps of what an HCT setup might entail, and have included the list in Appendix A12. We also provide a brief description of the steps on page 5 of the main manuscript.

11. Epi model section line 211: what is being modeled? Throughout the manuscript, “infections” is used—symptomatic infections, which is the primary endpoint being used by current vaccine studies?

Response:

We were referring to the number of confirmed COVID-19 cases, similar to what is reported by the Center for Systems Science and Engineering (CSSE) at Johns Hopkins University, or reported by official sources. We now make this explicit by rephrasing line 266 to read “We forecast the cumulative number of confirmed infections and confirmed deaths in each state between February 29, 2020, and December 31, 2022, using the SIRDCV described by Eq. A.32 to Eq. A.38 in the Appendix before summing over all states in order to produce estimates for the entire U.S.”

12. Table 1: what exactly does “superiority by margin of 50%” mean? FDA has specified that the confidence interval should lie above 30%.

Response:

Thank you for catching this. We have now updated Appendix A11 to include results for superiority-by-margin of 30% (i.e., H0efficacy: VE ≤ 0.3, the same statistical hypotheses used in the Moderna study protocol2).

13. Eq A16—shouldn’t those lambdas be lower case?

Response:

We have checked that the equation is correct. In the equation, 𝜆 is defined as the force of infection (expected number of new cases of the disease per unit person-time at risk), and 𝛬 is defined as the cumulative force of infection (integral of the force of infection 𝜆 over the duration of the surveillance period). When the risk of infection is small, the cumulative force of infection 𝛬 is approximately equal to the risk of infection.7 In addition, under the proportional hazards assumption, Equation A.16 can be re-written as Equation A.17.

14. P. 4 Appendix—use of Pocock’s—some companies are using this, others versions of O’Brien-Fleming, which may lengthen the trial—should at least say something in the Discussion about this.

Response:

Thank you for suggesting this. We have now added a note on page 5 to highlight the differences between the Pocock’s test used in our simulations and the O’Brien & Fleming’s test used by some vaccine companies.

15. Although 504 configurations are referred to, I did not see it spelled out anywhere just what these are. Don’t make the reader work to figure out stuff like this.

Response:

Thank you for the suggestion. We have now updated Table 1 so that all the configurations simulated in our analysis are described clearly:

Parameter Values Number of combinations

Trial design RCT, ORCT, ARCT, HCT (30-day set-up), HCT (60-day set-up), HCT (90-day set-up), HCT (120-day set-up) 7

Vaccine efficacy of hypothetical candidate (%) 30, 50, 70, 90 4

Efficacy requirement Superiority, superiority by margin of 30%, superiority by margin of 50% 3

Epidemiological scenario Status quo, ramp, behavioral 3

Population vaccination schedule (doses/day) 1M, 10M, infinite 3

The total number of configurations simulated can be computed as the product of the last column: 7×4×3×3×3=756.

16. I couldn’t even tell how much the eventual attack rates varied, which is critical to the time course. Baseline numbers are from all 50 states, but recruitment is highly varied geographically, and some trials are trying to recruit high-risk individuals—need to know how much these rates are varying so we can see the effect of eventual attack patterns on timelines.

Response:

The attack rates are based on the simulated models and the parameters for the models are reported in Table A3 in the appendix. The attack rates are dependent on the individual states’ initial conditions and the assumptions made, and can only be obtained by feeding the parameters into the models, which we will open-source when the manuscript is accepted.

We acknowledge that companies will try to recruit high risk individuals in places where there are high rates of infections. We had to make the assumption that the trials are distributed over the entire U.S. because exact details of the trials are not available at the time of writing. Furthermore, the companies modify their plans continuously because of the rapidly changing epidemic situation and it is difficult to predict how they will do it. Finally, what we are proposing is a general framework to compute the cost-benefit analysis. We want to keep the framework general so that other groups can reuse it to simulate the attack rates based on their own assumptions and/or models.

17. Unfortunately, a great deal of effort has gone into the extensive simulations. We could have had the same results without any simulations—just tabulate a wide range of approximate final attack rates, and show when how many cases would be accumulated under each, and then do the cost-benefit analyses on those. Other groups have made much more extensive simulations of the pandemic; the focus of this manuscript is not coming up with accurate trajectories.

Response:

In this paper, we seek a general framework for the computation of the cost-benefit of various vaccine trial designs. While we could have assumed an attack rate for the duration of the clinical trial, we think that it is more general and realistic to assume a set of epidemiology models to simulate the outcomes. We want to keep the framework general so that other groups that have made more extensive simulations of the pandemic can plug in their models and reuse our framework to simulate the attack rates.

Reviewer 2

This manuscript is well written and informs the statistical rationale for evaluating the common three clinical trial designs against less common (HCT) to be considered for phase III COVD-19 clinical vaccine trials. The designs described are the traditional vaccine efficacy randomized clinical trial (RCT), optimized vaccine efficacy RCT (ORCT) where surveillance period is optimized, adaptive vaccine efficacy RCT (ARCT) with sequential analysis of data, and human challenge trial (HCT) with advantages of smaller participant size and less monitoring time but with more health risk and more set up time. These designs are clearly characterized with minimal biases.

1. However, the authors did not integrate the published information already available before their submission on the four COVID-19 vaccine trials which are ongoing world-wide and supported and/or monitored by NIH. These are the vaccine trials by Moderna, AstraZeneka, Pfizer/BioNTech, and possibly Johnson and Johnson’s Janssen Pharmaceutical. However, the manuscript cites the website announcements of Moderna, AstraZeneka, and Pfizer/BioNTech.

Instead of only citing the company’s website announcements, the authors should have included in their manuscript Moderna’s preliminary report on their phase 1 dose-escalation, open-label trial in New England Journals of Medicine (published on July 14, 2020; DOI: 10.1056/NEJMoa2022483) and AstroZeneka’s phase 1/2 single-blind RCT in Lancet (published online on July 20, 2020; https://doi.org/10.1016/ S0140-6736(20)31604-4). If the authors have read these articles, they would have realized the level of reactogenicity or adverse effects that these two vaccines have in their phase I trials. The AstroZeneka’s vaccine, compared to the licensed meningococcal vaccine control group, had more local and systemic adverse reactions which lasted up to seven days. Both if these phase I trials used healthy 18-55 years old but the severity of such adverse effects is unknown to those who have comorbidity and to older population at the time of manuscript submission. The authors should describe the potential consequences of high adverse vaccine effect. It appears that none of the formula included in the appendix integrates such a risk factor. The risk factor that either the vaccine delivery system used (ChAdOx1 vector, mRNA-LNP, Ad26 vector) and/or the vaccine immunogen expression may have caused higher adverse effect which may also impact the accrual of the participants.

Response:

This is an excellent point. We have now added references to Moderna’s and AstraZeneca’s articles on page 8. We share the reviewer’s concern about the potential adverse effects of the vaccine candidates. It is without a doubt that the safety of a vaccine to be given to the masses should be of utmost priority in clinical trial development. However, we hesitate to describe any highly-adverse potential side effects of a COVID-19 vaccine for several reasons. First, there is no evidence of serious adverse effects for current vaccine candidates. The two Phase 1 studies cited by the reviewer reported that majority of the adverse events in the treatment arm were only mild and moderate in severity, with no serious adverse reactions observed.

Due to the lack of expertise and access to the actual clinical data, we are also reluctant to speculate on the tolerability of the vaccine in patient populations not studied in the Phase 1 trials (e.g., the older population and those with comorbidities), to avoid adding to the misinformation regarding vaccination. We believe that such issues are best addressed by vaccine experts, who will have the most accurate information. Lastly, we believe that vaccine candidates authorized for large-scale Phase 3 efficacy trials should be generally safe, otherwise they would have been halted by the FDA in early-stage clinical testing. Therefore, we assume in our simulations that the hypothetical vaccine candidate is generally well tolerated without any serious side effects. We explicitly acknowledge this assumption on page 8 of the manuscript.

2. More importantly, authors should describe what kind of immunity must be generated by the COVID vaccine to be effective in humans since vaccine efficacy should also be based on immune correlate of vaccine protection.

Response:

The immunity required to prevent COVID-19 is an important topic that is beyond the scope of this study, which focuses on the costs/benefits of different vaccine efficacy clinical trial designs, as opposed to the biological mechanisms of vaccination. That being said, SARS-CoV-2 is a novel pathogen for which there is little information regarding the immune response (i.e., level of antibody) required for immunity (i.e., threshold of protection). As a result, vaccine efficacy can only be confirmed using the longer and more costly vaccine efficacy clinical trials, versus the shorter vaccine immunogenicity trials where correlates of protection are used as surrogate endpoints. We explain this on page 2 of the main manuscript.

3. The phase III trials are all in place for the four companies at the time of manuscript review but should not be required for this manuscript since such information although on Clinicaltrials.gov were confirmed by the companies based on the results from phase I or 1/2 trials which were after the manuscript submission. Nevertheless, none of the four phase II trials uses HCT. In addition, FDA and NIH have not declared in peer-reviewed journal publication(s) that HCT is currently being considered for COVID vaccine. Please do not use citation such as in reference 44 (politico from White House). Please use peer reviewed journal citation for such a controversial subject.

Response:

Thank you for pointing this out. On page 11 of the manuscript, we mention that the NIAID appears to have made some headway in developing a challenge strain. This is consistent with statements by the NIAID.8,9 (We did not make any comments regarding the FDA/NIH considering/eliminating the possibility of using HCTs for COVID-19 vaccine development.) Unfortunately, we did not find any peer-reviewed journal publications by the FDA/NIH regarding the use of HCTs. However, we have seen several reports citing Dr. Anthony Fauci referring to challenge trials as the NIAID’s “Plan D.”8

4. In light of not knowing whether CoV-2 spike protein can be safely administered as vaccine immunogen without long-term adverse incident, the use of HCT is a concern. Until safety of the volunteers can be fully met with effective therapy against COVID-19, HCT may not be a consideration for prophylactic COVID-19 vaccine trial. An example of another emerging pathogen that have undergone extensive clinical vaccine trials is the prophylactic HIV-1 vaccines, but HCT has yet to be used. Instead, newer HIV-1 vaccine designs are tested in HIV-1 positive individuals as therapeutic vaccine, which is a more scientifically and ethically rationale approach.

Response:

This point is well taken. We agree that the lack of an effective therapy against COVID-19 is a major obstacle for challenge trials. However, we believe that the recent approval of remdesivir by the FDA as a therapeutic for COVID-1910 can alleviate at least some of the concerns, thus potentially paving the way for an HCT (e.g., the UK government has announced their intent to conduct human challenge studies for COVID-1911).

5. The transmission rate of the SARS-CoV-2 is rapid without mask and appropriate distancing. Hence consideration should be in how you recruit the volunteers such as those who wear mask versus those who do not wear mask.

Response:

Thank you for the suggestion. From our review, mask-wearing behaviors do not seem to be part of the inclusion/exclusion criteria of current vaccine trials.1,2 In addition, without access to data to inform our assumptions and models, we believe that introducing such components to the simulation framework only adds unnecessary complexity and may provide a false sense of precision. Therefore, we have decided not to model mask-wearing behaviors in our simulations.

Overall, the authors are raising an interesting question of the potential of using HCT based on how much lives saved and level of morbidity decreased which is an interesting subject to be raised to the scientific community without biases but with care.

 

References

1. Pfizer. Coronavirus COVID-19 Scientific Research and Resources. https://www.pfizer.com/science/coronavirus (2020).

2. Moderna. COVE Study: Participate to Make a World of Difference. https://www.modernatx.com/cove-study (2020).

3. Chris Jennison, B. T. Group Sequential Designs for Survival Data. in Handbook of Survival Analysis (eds. Klein, J. P., van Houwelingen, H. C., Ibrahim, J. G. & Scheike, T. H.) 595 (CRC Press, 2013).

4. Stallard, N. et al. Efficient Adaptive Designs for Clinical Trials of Interventions for COVID-19. Stat. Biopharm. Res. (2020) doi:10.1080/19466315.2020.1790415.

5. Chaudhuri, S., Lo, A. W., Xiao, D. & Xu, Q. Bayesian Adaptive Clinical Trials for Anti-Infective Therapeutics During Epidemic Outbreaks. Harvard Data Sci. Rev. (2020) doi:10.1162/99608f92.7656c213.

6. Pallmann, P. et al. Adaptive designs in clinical trials: Why use them, and how to run and report them. BMC Med. 16, 1–15 (2018).

7. Nauta, J. Statistics in Clinical and Observational Vaccine Studies. (Springer International Publishing, 2020). doi:10.1007/978-3-030-37693-2.

8. Gupta, S., Cohen, E. & Howard, J. Covid-19 vaccine research: US scientists considering coronavirus strain for potential human challenge trials as ‘Plan D’. https://www.cnn.com/2020/08/14/health/coronavirus-vaccine-strain-us-scientists-bn/index.html (2020).

9. Steenhuysen, J. U.S. to make coronavirus strain for possible human challenge trials | Reuters. https://www.reuters.com/article/us-health-coronavirus-vaccine-challenge/exclusive-u-s-to-make-coronavirus-strain-for-possible-human-challenge-trials-idUSKCN25A1EL (2020).

10. FDA. FDA Approves First Treatment for COVID-19. https://www.fda.gov/news-events/press-announcements/fda-approves-first-treatment-covid-19 (2020).

11. Kirby, T. COVID-19 human challenge studies in the UK. Lancet Respir. Med. (2020) doi:10.1016/S2213-2600(20)30518-X.

---

## [Editor Report · Decision Letter 1]

10 Dec 2020

A cost/benefit analysis of clinical trial designs for COVID-19 vaccine candidates

PONE-D-20-25637R1

Dear Dr. Lo,

We’re pleased to inform you that your manuscript has been judged scientifically suitable for publication and will be formally accepted for publication once it meets all outstanding technical requirements.

Kind regards,

Xia Jin, MD, PhD

Academic Editor

PLOS ONE
---

## [Editor Report · Acceptance letter]

15 Dec 2020

PONE-D-20-25637R1 

A cost/benefit analysis of clinical trial designs for COVID-19 vaccine candidates 

Dear Dr. Lo:

I'm pleased to inform you that your manuscript has been deemed suitable for publication in PLOS ONE. Congratulations! Your manuscript is now with our production department. 

Kind regards, 

on behalf of

Dr. Xia Jin 

Academic Editor

PLOS ONE